# Immunogenicity and Safety of Half-Dose Heterologous mRNA-1273 Booster Vaccination for Adults Primed with the CoronaVac^®^ and ChAdOx1-S Vaccines for SARS-CoV-2

**DOI:** 10.3390/vaccines12040344

**Published:** 2024-03-22

**Authors:** Nina Dwi Putri, Aqila Sakina Zhafira, Pratama Wicaksana, Robert Sinto, Gryselda Hanafi, Lowilius Wiyono, Ari Prayitno, Mulya Rahma Karyanti, Murni Luciana Naibaho, Febrina Febrina, Hadyana Sukandar, Vivi Setiawaty, Mursinah Mursinah, Ahmat Rediansya Putra, Heri Wibowo, Julitasari Sundoro, Hindra Irawan Satari, Dwi Oktavia, Pretty Multihartina, Dante Saksono Harbuwono, Sri Rezeki Hadinegoro

**Affiliations:** 1Faculty of Medicine, Universitas Indonesia—Cipto Mangunkusumo Hospital, Jakarta 10430, Indonesia; aqilasz9@gmail.com (A.S.Z.); pratamawicaksana@yahoo.com (P.W.); rsinto@yahoo.com (R.S.); gryseldahanafi@gmail.com (G.H.); lowilius.wiyono@alumni.ui.ac.id (L.W.); ariprayitno@ikafkui.net (A.P.); karyanti@ikafkui.net (M.R.K.); hadyanas@yahoo.com (H.S.); hsatari@ikafkui.net (H.I.S.); shadinegoro46@gmail.com (S.R.H.); 2Cempaka Putih Public Health Center, Jakarta 10520, Indonesia; murninaibaho20@gmail.com (M.L.N.); bangunfebrina@yahoo.com (F.F.); 3National Institute of Health Research & Development, Jakarta 14530, Indonesia; vivisetiawaty@hotmail.com (V.S.); mursinah_my@yahoo.com (M.M.); pmdsasono@gmail.com (P.M.); 4Diagnostic and Research Center, Faculty of Medicine, Universitas Indonesia, Jakarta 10430, Indonesia; amekredi16@gmail.com (A.R.P.); bowoheri04@gmail.com (H.W.); 5The Indonesian Technical Advisory Group on Immunization, Jakarta 10430, Indonesia; julitasari.sundoro@gmail.com; 6Jakarta Health Agency, Jakarta 10160, Indonesia; dokterlies@yahoo.com; 7Ministry of Health of Republic of Indonesia, Jakarta 12750, Indonesia; danteid@yahoo.com

**Keywords:** COVID-19, vaccine, half-dose, mRNA, booster, safety, efficacy

## Abstract

Coronavirus disease 2019 (COVID-19) has been extensively researched, particularly with regard to COVID-19 vaccines. However, issues with logistics and availability might cause delays in vaccination programs. Thus, the efficacy and safety of half-dose heterologous mRNA should be explored. This was an open-label observational study to evaluate the immunogenicity and safety of half-dose mRNA-1273 as a booster vaccine among adults aged >18 years who underwent a complete primary SARS-CoV-2 (severe acute respiratory syndrome coronavirus 2) vaccination regimen with CoronaVac^®^ and ChAdOx1-S. Adverse events (AEs), seropositivity rate, seroconversion, geometric mean titer (GMT) of SARS-CoV-2 antibodies, neutralizing antibodies, and T cells (CD4+ and CD8+) specific for SARS-CoV-2 were analyzed. Two hundred subjects were included in the final analysis, with 100 subjects in each priming vaccine group. Most of the AEs were mild, with systemic manifestations occurring between 1 and 7 days following vaccination. A significant difference was observed in the GMT and seropositivity rate following booster dose administration between the two groups. CD8+/CD3+, IFN (interferon)-producing CD8+, and TNF (tumor necrosis factor)-producing CD8+ cells showed significant increases in both groups. The administration of the half-dose mRNA-1273 booster is safe and effective in increasing protection against SARS-CoV-2 infection.

## 1. Introduction

Coronavirus disease 2019 (COVID-19) vaccines have been extensively studied in recent years to prevent the morbidity and mortality caused by SARS-CoV-2 infections. Countries with a high vaccination coverage, including some in the Middle East, initiated boosting programs with heterologous vaccines even before a solid scientific basis for the procedure was published [1]. Recent evidence suggests that heterologous messenger RNA (mRNA) boosters are highly immunogenic and relatively safe [2,3]. The Moderna COVID-19 vaccine (mRNA-1273) was approved during the initial phase of emergency-use authorization in 87 countries as the primary vaccine and is being used as a booster vaccine in several countries, including Indonesia [4]. Numerous reports have signified that mRNA-based vaccines tend to elicit greater local and systemic responses compared to other COVID-19 vaccine types [5,6,7]. 

Indonesia is among the countries that kicked off its vaccination program by supporting the COVID-19 Vaccines Global Access (COVAX) initiative, first vaccinating frontline healthcare workers [8]. When the number of positive cases was at its highest and the morbidity rates among healthcare workers rose, the Indonesian government decided to attempt to boost immunity by providing booster vaccinations for this group [9]. Half-dose mRNA-1273 was shown to induce robust antibody responses and be well tolerated and safe in healthy adults [10]. Using only half a dose for boosting could save significant quantities of vaccines and consequently increase the number of people who can receive the vaccine.

Lower vaccine doses (fractionations) were previously used during outbreaks of yellow fever and polio in 2016 to maximize vaccine supply. The WHO (World Health Organization) advised using one-fifth of the regular dose of the yellow fever vaccine amid concerns about the limited supply in Africa [10]. The recommendation was based on several reports that a lower dose exhibited immunogenicity comparable to that of its full-dose counterpart [11,12]. The present study was conducted to investigate the efficacy of a half-dose booster in inducing an adequate immune response while minimizing side effects in vaccinated people and, therefore, provide a solution to the issue of global COVID-19 vaccine supply constraints. This study might generate essential data on the merits of vaccinating people with half of the standard dose of the mRNA-1273 COVID-19 vaccine.

## 2. Materials and Methods

### 2.1. Investigational Products

The Moderna mRNA-1273 COVID-19 vaccine is a lipid nanoparticle-encapsulated mRNA vaccine that expresses the prefusion-stabilized spike glycoprotein [13]. Multiple-dose vials of 5.5 mL and 7.5 mL were available for this study. The products were imported and stored frozen between −50 °C and −15 °C. Subsequently, the vaccines were transported to the investigation site with minimal shaking and vibration while ensuring the temperature remained constant at 2–8 °C until usage. Vials were stored between 2 °C and 25 °C during usage and discarded 12 h after the first use. The booster dose was administered intramuscularly in the subjects’ left deltoid muscle (unless contraindicated), with each dose containing 50 µg, by a trained vaccinator team.

### 2.2. Study Design, Ethics, and Blinding

This was an open-label observational study to evaluate the immunogenicity and safety of half-dose mRNA-1273 as a booster vaccine among adults fully vaccinated with ChAdOx1-S (Oxford/AstraZeneca; UK) and CoronaVac^®^ (Sinovac; Beijing, China). No blinding of subjects or investigators took place during the intervention (booster injection). However, the laboratory team that handled the specimens was blinded to the subjects’ priming group. No placebo group was included as a comparator in this study. This study was conducted according to the latest revision of the Declaration of Helsinki, International Conference on Harmonization—Good Clinical Practice guidelines and local regulatory requirements. This study was also approved by the local ethics committee (approval number LB.02.01/2/KE.014/2022).

### 2.3. Trial Protocol

Adults aged >18 years who completed a primary vaccination series against SARS-CoV-2 with CoronaVac^®^ 6 months prior and ChAdOx1-S6 9 months prior to the time of enrollment were enrolled in this study. Those who fulfilled the inclusion criteria were given in-depth explanations of this study’s objectives, methods, risks, and benefits before they signed a written informed consent form. 

The exclusion criteria were as follows: (1) inoculation with a third dose of a SARS-CoV-2 vaccine; (2) participation or plans to participate in other clinical trials; (3) complaints of fever (defined as a body temperature of >37.5 °C measured with an infrared thermometer/thermal gun), upper respiratory tract symptoms (sneezing, nasal congestion, a runny nose, a cough, a sore throat, ageusia, chills, or shortness of breath) within 72 h before enrollment; (4) a blood pressure >180/110 mmHg; (5) a confirmed history of COVID-19 within 1 month of study commencement (laboratory confirmation); (6) a history of serious adverse reactions to any vaccines or vaccine ingredients; (7) known comorbidities including uncontrolled autoimmune disease, a history of uncontrolled coagulopathy or blood disorders, immune deficiency, receipt of a blood-derived product or transfusion within 3 months of enrollment, immunosuppressant therapy, uncontrolled chronic disease, and a history of uncontrolled epilepsy within the last 2 years or other progressive neurological disorders; (8) receipt of any vaccination within 1 month before and after boosting with the study vaccine; (9) pregnancy; and (10) age >60 years, with complaints of heart failure or at least five concurrent comorbidities (hypertension, diabetes, cancer, chronic lung disease, heart attack, congestive heart failure, chest pain, asthma, joint pain, stroke, and kidney disease). 

A total of three visits were conducted, with an additional phone call 24 h after vaccination. The flowchart of the study procedure is shown in Figure 1. The first visit (V1) consisted of a physical examination, urine pregnancy test for female subjects of childbearing age, collection of 10 mL pre-vaccination blood samples, and administration of the mRNA-1273 booster dose vaccine. Subjects were observed for 30 min for immediate adverse events before discharge. Twenty-four hours after vaccination, the surveillance team called all subjects regarding any adverse events, administering a structured questionnaire. The second visit (V2) was conducted on day 7 following the trial. The final visit (V3) took place 28 days after vaccination. Data on physical examination, recent medical history, serious adverse events, and COVID-19 cases were collected. Another 10 mL blood sample was obtained for immunogenicity tests. All reactogenicity data, COVID-19 infections, and concomitant therapy were recorded throughout the study period.

A nasopharyngeal swab sample was collected from study participants showing symptoms consistent with COVID-19, including a cough, taste or smell disorders, dyspnea, fever, chills, a sore throat, fatigue, nasal congestion or a runny nose, body pain, muscle pain, headache, nausea, vomiting, or diarrhea for at least two consecutive days. Samples were collected with swab sticks and placed into a viral transport medium (VTM), stored in a cooler box at 2–8 °C, and transported to the designated laboratory. Subjects were monitored until recovery. Clinical data related to the illness were recorded for both hospitalized and outpatient subjects. Subjects were compensated for their travel expenses (IDR 200,000 [approximately USD 13.3] per visit), and treatment for any adverse events occurring throughout the study period was to be paid for by the investigators.

### 2.4. Primary and Secondary Outcomes

Reactogenicity was evaluated by both subjects and investigators and recorded at 24 h, 7 days, and 28 days following vaccine administration. The 24 h evaluation was conducted via a phone call, while other time points were assessed via interview and diary card records. All reactogenicity data were classified into four grades according to the Food and Drug Administration toxicity grading scale [14]. Any signs, symptoms, or disease, including unfavorable laboratory findings occurring after administration of the investigational product, whether considered as related or unrelated to this study, were recorded. Solicited adverse events actively observed within 7 days after the booster dose vaccination included local (pain, tenderness, erythematous skin, and induration) and systemic (fever, nausea, vomiting, headache, fatigue, malaise, myalgia, arthralgia, syncope, and chest pain) reactions [13]. Any deaths, life-threatening medical conditions, hospitalization, disability or incapacity, and congenital anomaly or birth defects after receiving the vaccines were recorded as serious adverse events (SAEs) based on the Good Clinical Practice (GCP) guidelines.

Seropositivity rate, seroconversion, geometric mean titer (GMT) of SARS-CoV-2 antibodies, neutralizing antibodies, and CD4+ and CD8+ T cells to SARS-CoV-2 at 28 days were recorded as the secondary evaluation criteria. Ten milliliters of blood was collected in a BD Vacutainer^®^ tube during V1 and V3 for immunogenicity examination. Samples for anti-SARS-CoV-2 and neutralizing antibody tests were collected in an anticoagulant Vacutainer^®^ tube. Blood samples were allowed to clot for 30 min-to-2 h at room temperature and centrifuged at 3000 rpm for 15 min. Following this, the blood samples were refrigerated at 2–8 °C after the clotting period and centrifuged within 24 h after collection. One aliquot of 1 mL was used for electrochemiluminescence immunoassay (ECLIA) for SARS-CoV-2 antibody analysis, and four aliquots of 4 mL were used for neutralizing antibody analysis with the CPass™ Surrogate Virus Neutralization Test (sVNT) Kit at the National Institute of Health Research and Development (NIHRD) Laboratory. The ECLIA and sVNT cut-offs for positive serology results were 0.4 U/mL and 30%, respectively. The sVNT was achieved from the inhibition formula of 1 − (optical density of sample/optical density of negative control) × 100%; a larger sVNT value indicated poor seropositivity. Plaque-reduction neutralization tests (PRNTs) were performed with the original SARS-CoV-2 Wuhan strain and the Delta strain. The test comprised four parts: virus back titration; serum preparation and dilution; neutralization; and fixation and coloring. Inactivated serum was processed at 56 °C for 30 min.

The gating strategy used in the intracellular cytokine staining (ICS) was as follows: CD3+ lymphocytes were classified into CD4+ and CD8+. Each cell was then classified based on CD154+, along with TNF and IFN results (Appendix A).

### 2.5. Statistical Analyses

The sample size for immunogenicity analysis was calculated using the difference of two proportions with the assumption that this study would show at least 95% seropositivity, a level of significance of 5%, and a power of 80%. The reactogenicity analysis was determined based on the 69.8% incidence rate of adverse events from a previously published study in Bandung, Indonesia [15], with the sample size calculated using the formula for proportion within a population, a confidence interval of 95%, and a precision of 10%. Based on these calculations, the sample size required was 100 subjects per group. 

The *t*-test was used to compare the differences in the proportion of groups with a primary dose of the CoronaVac^®^ vaccine and groups with a primary dose of the ChAdOx1-S vaccine. Analysis of variance (ANOVA) was performed to compare the differences in the mean GMT of ten groups with a primary dose of CoronaVac^®^ and five groups with a primary dose of the ChAdOx1-S vaccine. A post hoc test was performed for significant results with the least significant difference test. A *p*-value < 0.05 was considered significant.

## 3. Results

A total of 213 participants were screened, and 13 were excluded from this study. Thus, 200 subjects were included in the final analysis, with 100 subjects in each priming vaccine group. Table 1 shows the demographic data of the study participants. Most subjects were male (67.0%), with a mean age of 40.9 ± 11.64 years old. The duration since the last vaccination was approximately 6–8 months.

### 3.1. Reactogenicity of Half-Dose mRNA Vaccine Booster

Table 2 shows the reactogenicity of the subjects at 24 h, 7 days, and 8 days after vaccination. Most of the AEs were mild (76.7%), with systemic manifestations (56.3%) occurring between 1 and 7 days following vaccination. Most subjects recovered from the AEs (99.7%), with only one subject not recovering from a persistent cough after 28 days of observation. One grade 3 AE each occurred in the CoronaVac^®^ group and the ChAdOx1-s group. Both cases occurred within 1–7 days of vaccine administration and consisted of a high fever (>40 °C), but the subjects recovered after treatment within a day. Unsolicited local AEs occurred only in one subject in the CoronaVac^®^ priming group, while unsolicited systemic adverse events occurred in 23 subjects (13.7%). The identified systemic adverse events included coughing (six subjects), fever (four subjects), toothache (three subjects), diarrhea (two subjects), and increased blood pressure (two subjects). In addition, one subject each complained of a stuffy/runny nose, red eyes, lower back pain, and back pain and vertigo following vaccination. No serious adverse event was recorded in this study for both groups.

### 3.2. Immunogenicity Evaluation (Anti-sRBD [Spike-Receptor Binding Domain] IgG)

The anti-sRBD IgG level was evaluated using the GMT, seropositivity rate, seroconversion, and geometric mean increase (GMI). On comparing the pre- and post-booster levels based on the vaccine priming, on day 28 after vaccination both the GMT and the seropositivity rate showed no statistical difference between the groups; however, the group with the CoronaVac^®^ prime showed a significantly higher seroconversion rate (56 vs. 36, *p* = 0.045) and GMI (15.32 vs. 5.16, *p* < 0.001; Table 3). However, it should be noted that a significantly lower GMT and seropositivity rate were found in the CoronaVac^®^ group prior to booster administration, which might explain the trend of higher seroconversion and GMI (Figure 2).

Figure 3 compares the sVNT inhibition capacity of ChAdOx1-S and CoronaVac^®^. No significant difference was found in the inhibition capacity measured with the sVNT antibody assay before the booster dose was administered between the ChAdOX1-S and CoronaVac^®^ groups (*p* = 0.328). However, a significant difference was seen after the booster dose was administered between the two groups (*p* = 0.024). 

Table 4 shows the serological conversions for SARS-CoV-2 antibodies before and 28 days after boosting with half-dose mRNA-1273. As the data distribution was not normal, the data were presented as the median (IQR [interquartile range]). All priming groups showed a significant increase in SARS-CoV-2-neutralizing antibodies, measured with sVNT assays (*p* = 0.000). However, a significant difference was found between the two groups after booster vaccination, although the median value was the same at 0.96 (0.95–0.97). 

The PRNT results of eight subjects from each group are shown in Table 5. Overall, no significant difference in the PRNT between the ChAdOX1-S and CoronaVac^®^ groups before and after booster dose administration was observed. However, a significant increase in the neutralization capacity was observed for the Wuhan strain in both ChAdOX1-S (*p* = 0.012) and CoronaVac^®^ (*p* = 0.028) groups, which was not seen for the SARS-CoV-2 Delta strain.

### 3.3. Cell-Mediated Immunity (Specific T Cell Responses)

Regarding the T cell-mediated immune response to SARS-CoV-2, the CD4+ and CD8+ cell proportions, along with the proportions of AIM+ (activation-induced marker +) CD4+ cells, were recorded. Positive responses to spike antigenic stimulation occurred in the CD4+ or CD8+ compartment, with a significant increase in cytokine-producing cells among both CD4+ and CD8+ cells (Table 6 and Figure 4). In the CD4+ compartment, a significant increase was observed in IFN-producing cells and TNF-producing cells in both vaccine prime groups on day 28 of observation compared to the pre-booster status. However, no significant difference in IFN-producing cells (*p* = 1.00) on day 28 was observed despite the significant differences between the vaccine primes in the pre-booster analysis, with a higher proportion of IFN-producing CD4+ cells in the CoronaVac^®^ group. The CD4+/CD3+ proportion was also found to be significantly lower at the V3 observation compared to V1 in both the ChAdOX1-S (*p* < 0.001) and CoronaVac^®^ (*p* = 0.001) groups, with a further decline in the CoronaVac^®^ group compared to the ChAdOX1S group (*p* = 0.002). 

The observation of CD8+/CD3+ cells revealed a significant increase in both the ChAdOX1-S group (*p* = 0.001) and the CoronaVac^®^ group (*p* = 0.002). A similar trend was also observed for IFN-producing CD8+ cells and TNF-producing CD8+ cells (*p* < 0.001). However, at the V3 observation the CoronaVac^®^ vaccine group had a significantly higher proportion of IFN-producing (*p* = 0.035) and TNF-producing (*p* = 0.001) CD8+ cells.

## 4. Discussion

Since the first report of COVID-19 in November 2019, various scientific organizations have joined hands to develop studies on its treatment and prevention. As of 12 January 2022, nine vaccines have been approved by the WHO Emergency Use Listing Procedure, with research on many more candidates ongoing [16]. Indonesia initially approved the use of COVID-19 vaccines on 11 January 2021, with CoronaVac^®^ (Sinovac Biotech, China) being the first vaccine authorized by the Indonesian Food and Drug Control Agency (BPOM) [17]. ChAdOx1-S followed soon afterward, gaining approval by the Indonesian BPOM on 22 February 2021 [18]. Meanwhile, the first mRNA vaccine to be approved in Indonesia, mRNA-1273, was approved only 4 months after CoronaVac^®^.

The original COVID-19 vaccination regimen mostly consists of two doses, as one dose produces a relatively weak immune response [19,20]. Previous studies have shown that after 6 months of vaccination, neutralizing antibody titers decline substantially [21]. This raises the question of when to administer additional booster doses to maintain adequate immunogenicity. In addition, with the increasing number of Omicron-variant infections among Indonesian citizens, studies on the safety and efficacy of vaccines have become a hot topic [22]. T cells induced by the SARS-CoV-2 vaccine responded to the spike protein as a whole and could remain effective with few mutations [23]. Numerous studies have shown the benefit of booster dose administration, although the types and timing vary according to the available vaccines in the region [24]. A preliminary report in the United States showed that both homologous and heterologous booster vaccinations (mRNA-1273, Ad26.CoV2.S, and BNT162b2) were well tolerated and provided good immunity against SARS-CoV-2 [23]. 

The median duration from the second vaccination to the booster dose in this study was 190 days for ChAdOx1-S and 263 days for CoronaVac^®^. This difference could be explained by the discrepancy in the first approval date of these vaccines by the Indonesian BPOM. In addition, the booster dose was only approved in July 2021, with mRNA-1273 as the chosen vaccine, given to healthcare workers on priority [18]. Nevertheless, the two priming groups had well surpassed their predicted immunity-waning period of approximately 6 months.

Reactogenicity is defined as a variety of reactions that might occur due to increased immune responses after vaccination. Antigens in the vaccine typically induce the innate immune response through recognition by various receptors found on the local and peripheral immune cells circulating in the body. Following vaccine administration, the stimulation of the host immune response initiates a complex series of innate immune reactions that are very important to trigger strong antigen-specific acquired immune responses for protection against diseases [25]. 

All the observed AEs were comparable between the two groups. More than 75% of the AEs in this study were of grade 1 intensity, with tenderness being the most common local AE (71.8%) and myalgia (31.9%) the most common systemic AE. This finding was in accordance with the solicited local and systemic AEs reported between 0 and 7 days after vaccination by the Centers for Disease Control and Prevention (CDC) V-safe Surveillance System, which reported pain as the most common local AE (78.3%) [26]. Myalgia was the third most common AE for mRNA-1273 vaccination (51.4%), following fatigue (60.0%) and headache (53.2%) [26]. Most grade 2 AEs were classified based on the use of non-narcotic pain relievers for more than 24 h and mild disruption of daily activity. All grade 2 local AEs consisted of tenderness in the injection area. Moreover, most of the systemic AEs classified as grade 2 could be relieved by pain relievers. These included myalgia (26.5%), headache (14.7%), and joint pain (14.7%). All subjects except one recovered from the AEs; the exception complained of a continuing cough after 28 days following the booster vaccination. Two subjects with a grade 3 AE in the CoronaVac^®^ group and a grade 4 AE in the ChAdOx1-S group were so categorized due to a high fever (39.0–40.0 °C for grade 3 and >40 °C for grade 4). However, both recovered after treatment. This finding is similar to those of the CDC, where mRNA-1273 was shown to cause a grade 4-intensity fever of ≥40 °C in <0.1% of the first and second dose recipients [27]. 

Furthermore, 13.7% of subjects reported unsolicited systemic AEs. Respiratory symptoms such as coughing, a stuffy nose, and a runny nose are not typical mRNA-1273 AEs. Four of six subjects showed the symptoms of coughing between 8 and 28 days after booster vaccination, with 58.3% of them in the ChAdOX1-S priming group. These symptoms might be related to COVID-19 infection, although additional tests are preferable to determine the exact cause [28]. In addition, two subjects complained of grade 2 diarrhea, one from each priming group. According to the WHO, diarrhea is one of the typical side effects of COVID-19 vaccines and thus is classified as a solicited AE [29,30]. Moreover, a study on adverse reactions to the BNT162b2 and mRNA-1273 vaccines in Japan reported that 3–5% of the study participants had diarrhea after vaccination [31]. One subject from the ChADoX1-S group had both upper and lower back pain of grade 1 intensity, lasting 4–6 days, and recovered without any treatment. No prior studies reported back pain or lower back pain following mRNA-1273 vaccination. However, a BNT162b2 vaccine study in Saudi Arabia involving 455 subjects found that as high as 2.4% of the subjects complained of back pain after vaccination [32]. This might be caused by the systemic immune response to vaccination, which causes myalgia, although the exact reason it might affect the lower back region is not clear and requires further study.

Three of the CoronaVac^®^ priming group experienced toothache, all with grade 2 intensity. Previous reports revealed a very small number of subjects who experienced an orofacial AE with unknown mechanisms. The most common were Bell’s palsy, trigeminal neuralgia, and anaphylaxis-related symptoms [33,34]. Two subjects complained of increased blood pressure, one from the ChADoX1-S and the other from the CoronaVac^®^ priming group. One of the proposed underlying mechanisms is related to the “spike effect” after COVID-19 vaccination, where the imbalance between increasing angiotensin II and declining angiotensin 1–7 causes an acute elevation in blood pressure [35,36]. One subject in the CoronaVac^®^ group also experienced red eyes for 4 days with grade 1 intensity, between 8 and 28 days after vaccination. This incidence was not groundless, as several studies revealed that ocular manifestations of COVID-19 vaccination were possible, although the exact cause remains unknown [37,38,39]. Other observed reactions included vertigo in the ChADoX1-S priming group, experienced 1–7 days after booster dose vaccination. The GMT and seropositivity analysis of the sRBD-IgG also revealed a higher seroconversion rate and GMI in the CoronaVac^®^ group compared to the ChADoX1-S (AstraZeneca) group, with no significant differences of GMT and seropositivity rate 28 days after the booster vaccination. However, the significantly lower GMT and seropositivity found in the CoronaVac^®^ group prior to the booster dose vaccination should also be noted, which supports the need for booster vaccines in the CoronaVac^®^ priming group. This finding is supported by the work of Angkasekwinai et al., in Thailand, with a lower anti-SARS-CoV-2 sRBD IgG observed in the CoronaVac^®^ group compared to the ChADoX1-S group [40]. A similar post-booster finding, reported by Fadlyana et al., specifies a 100% seropositivity rate after the booster vaccine doses with both full-dose and half-dose use [15]. The findings support the possible use of half-dose booster vaccination rather than full-dose vaccination for better vaccine coverage where vaccine availability is limited [41].

The neutralizing antibody analyses were conducted with the sVNT assay. Neutralizing antibodies may be a good representation of the duration of protection against the disease, although they are not the only components considered in the determination of protectivity [42]. There was no significant difference in the inhibition capacity measured with the sVNT antibody assay before the booster administration between the ChAdOX1-S and CoronaVac^®^ groups (*p* = 0.328). This lack of difference contrasts with the number of positive proportions of seropositive subjects, where the CoronaVac^®^ group had a significantly lower positive sVNT proportion. This might be because subjects given mRNA vaccines had generally higher immune responses compared to those given inactivated vaccines. This difference in proportion could not be seen following booster dose administration. Interestingly, a significant difference in the median value of the inhibition percentage was found between the ChAdOX1-S and CoronaVac^®^ groups (*p* = 0.024). The CoronaVac^®^ subjects had a number of outliers, which may have made the discrepancy raise the statistical power. As this study was conducted with an intention-to-treat analysis, outliers were not removed. Thus, although there was a significant difference after vaccination, this would not cause any clinical difference in protection against the SARS-CoV-2 virus.

We also found comparable results in the ChAdOX1-S and CoronaVac^®^ groups regarding cell-mediated immunity. However, this study showed a higher proportion of %IFN/CD8+ and %NF/CD8+ in the CoronaVac^®^ group than in the ChAdOX1-S group. The results are in line with a previous study by Kang et al., on mRNA vaccination and cellular immunity: inactivated-virus vaccines such as CoronaVac^®^ elicited higher levels of cell-mediated immunity than mRNA vaccines due to the various virus parts and abundant immunogenic N protein increasing the Th1 response [43]. Previous reports found an increase in CD8+ and AIM+ CD4+ cells 18 months after SARS-CoV-2 infection. Th1 cells observed in individuals with COVID-19 demonstrated a distinctive expression of CXCR3 and CCR6, in addition to transcripts encoding cytokines and chemokines with anti-viral properties, such as CD154, IFN-γ, IL-2 (interleukin-2), and TNF (tumor necrosis factor) [44]. Previous findings show the majority of individuals who receive the vaccine are capable of generating a strong immune response from their CD4+ T cells for a duration of up to 6 months following vaccination. Additionally, a correlation is present between the initial CD4 T cell responses and the immune responses involving antibodies [45].

This study is not without limitations, particularly as we did not compare the efficacy and reactogenicity of half-dose vaccination with those of full-dose boosters. However, several studies support our rationale: previous studies in Indonesia revealed a non-significant difference between the half-dose and full-dose booster, with a comparatively lower adverse event rate [15]. Another compared the use of full-dose and half-dose boosters using BNT162b2 (Pfizer-BioNTech) with a similar seroresponse [46].

## 5. Conclusions

This study revealed that 28 days after mRNA-1273 booster administration, the neutralizing antibody titer and GMT of anti-sRBD IgG were comparable between the two priming groups (ChAdOx1-S and CoronaVac^®^). Administration of a half-dose mRNA-1273 booster was concluded to be safe and could provide good protection against SARS-CoV-2 infection.

## Figures and Tables

**Figure 1 vaccines-12-00344-f001:**
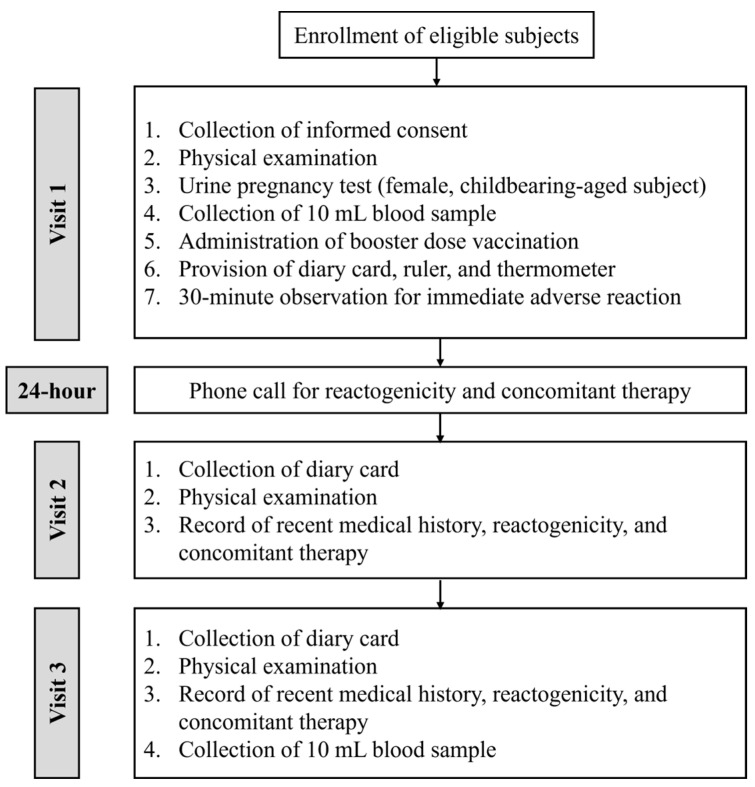
Flowchart of the study protocol.

**Figure 2 vaccines-12-00344-f002:**
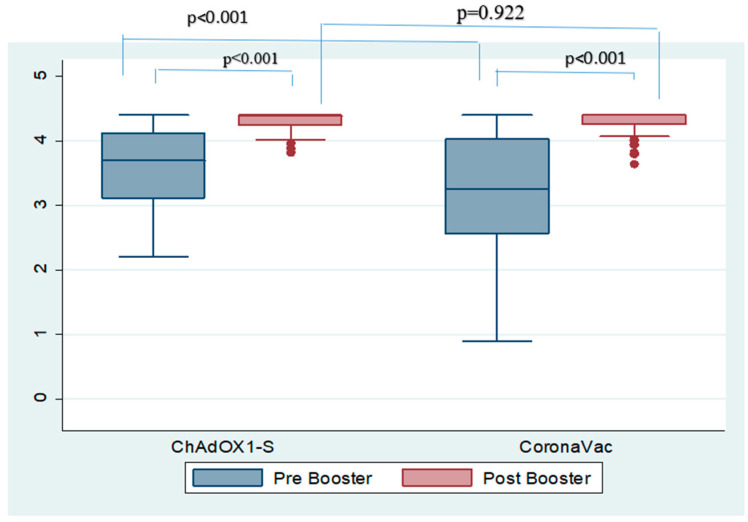
GMT antibodies before and after boosting with the Moderna COVID-19 vaccine.

**Figure 3 vaccines-12-00344-f003:**
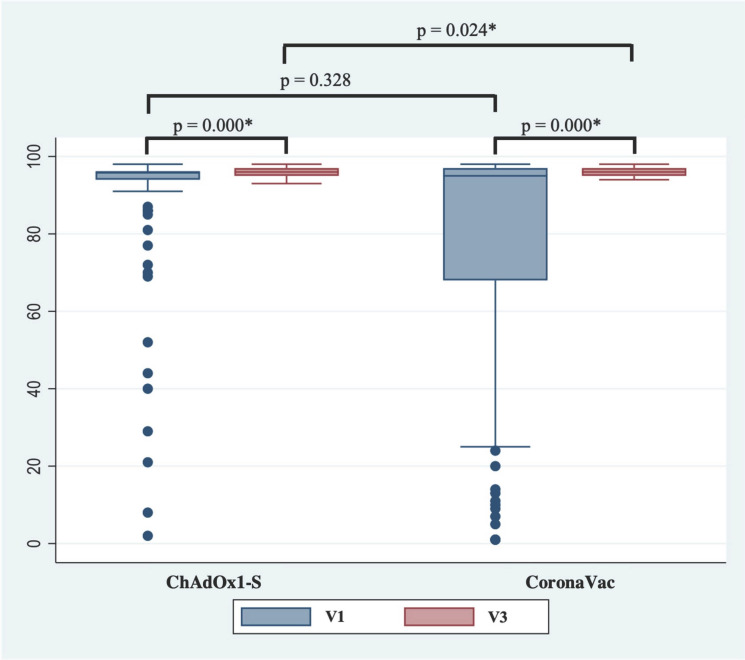
Inhibition capacity before and after booster mRNA-1273 vaccination. * a significant difference was observed (*p*-value < 0.05).

**Figure 4 vaccines-12-00344-f004:**
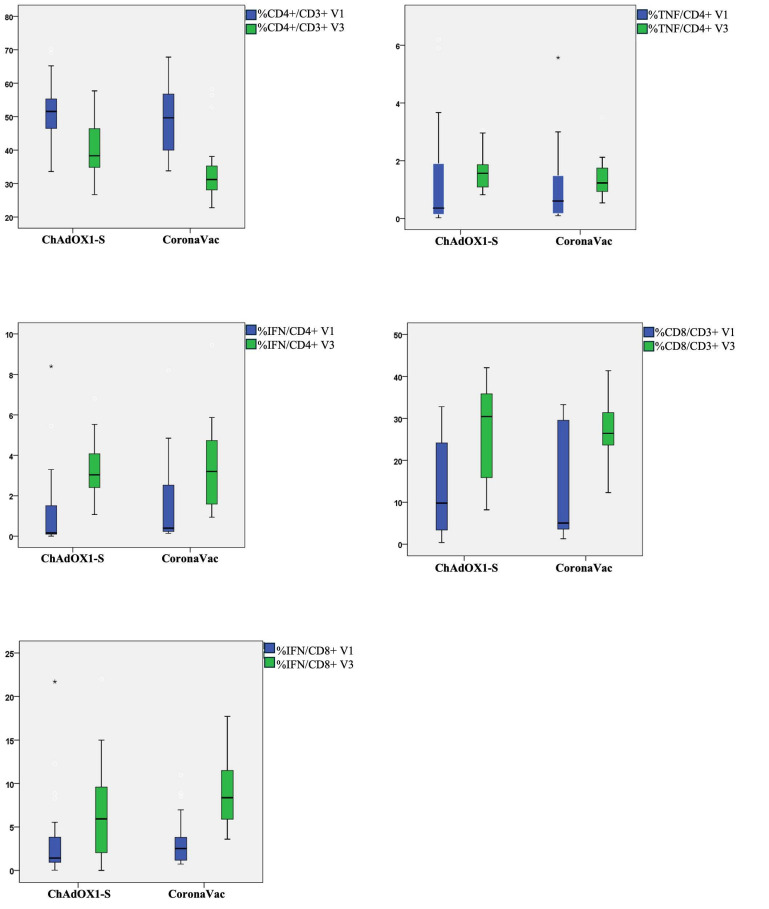
Cell-mediated immune response based on vaccine priming. * A significant difference was observed (*p*-value < 0.05).

**Table 1 vaccines-12-00344-t001:** Demographic data of study participants.

Variables	ChAdOx1-S	CoronaVac^®^
Male, *n* (%)	57 (57.0)	77 (77.0)
Age (years) 18–59 years ≥60 years	40.41 ± 1.32 98 (98.0) 11 (11.0)	41.57 ± 0.98 89 (89.0) 11 (11.0)
Days since last vaccination	190.0 (183.25–197.00)	263.0 (199.25–285.00)
Body weight (kg)	63.50 (57.05–72.63)	68.85 (58.13–83.48)
Body height (cm)	160.20 (154.85–166.68)	165.25 (159.00–169.15)

**Table 2 vaccines-12-00344-t002:** Distribution of adverse events.

Variables	ChAdOx1-S	CoronaVac^®^	*p*-Value
Types (*n* = 339)			0.866 ^a^
Local	68 (44.2)	80 (43.2)	
Systemic	86 (55.8)	105 (56.8)	
Duration since vaccination			0.985 ^b^
30 min	0 (0.0)	4 (2.2)	
1–7 days	142 (92.2)	171 (92.4)	
8–28 days	12 (7.8)	10 (5.4)	
Duration of adverse events	1 (1–2) days	1 (1–2) days	0.105 ^b^
Intensity			0.422 ^c^
Grade 1 (mild)	115 (74.7)	145 (78.4)	
Grade 2 (moderate)	38 (24.7)	39 (21.1)	
Grade 3 (severe)	1 (0.6)	1 (0.5)	
Grade 4 (potentially life-threatening)	0 (0.0)	0 (0.0)	
Local adverse events			N/A
Redness	2 (1.3)	4 (2.2)	
Tenderness	59 (38.3)	62 (33.5)	
Swelling	4 (2.6)	7 (3.8)	
Thickening of the skin	3 (1.9)	6 (3.2)	
Soreness	0 (0.0)	1 (0.5)	
Systemic adverse events			N/A
Fatigue	10 (6.5)	19 (10.3)	
Nausea/vomiting	4 (2.6)	7 (3.7)	
Myalgia	25 (16.2)	29 (15.7)	
Joint pain	12 (7.8)	14 (7.6)	
Headache	23 (14.9)	25 (13.5)	
Other systemic adverse events	12 (7.8)	11 (5.9)	

^a^ Chi-square test; ^b^ Mann–Whitney test; ^c^ Kruskal–Wallis test. ChAdOx1-S n = 154; CoronaVac^®^ *n* = 185. Bivariate analyses were not conducted for the types of adverse events.

**Table 3 vaccines-12-00344-t003:** Immunogenicity evaluation: IgG anti-sRBD titer before and after Moderna COVID-19 vaccine-boosting.

	Vaccine Priming		
	ChAdOx1-S (*n* = 100)	CoronaVac^®^ (*n* = 99)	*p*-Value *^)^
Pre-vaccination: GMT (U/mL) (95% CI) Median (Q1, Q3) Seropositive, n (%) (95% CI) 28 days after vaccination: GMT (U/mL) (95% CI) Median (Q1, Q3) Seropositive, n (%) (95% CI) Seroconversion, n (%) (95% CI) GMI (95% CI)	3848.57 (2930.89–5053.59) 4939 (1286–13,114.25) 100 (100) (96.38–100) 20,077.05 (28,940.87–21,281.39) 24,221.5 (17,164.5–25,000) 100 (100) (96.38–100) 46 (46) (35.98–56.26) 5.21 (3.97–6.85)	1375.62 (861.59–2196.34) 1806 (331.1–10,676.0) 89 (89.9) (82.21–95.05) 20,169.73 (18,745.63–21,702.01) 25,000 (17,920–25,000) 99 (100) (96.34–100) 61 (61.6) (51.30–71.22) 14.66 (9.32–23.07)	<0.001 0.001 0.922 1.000 0.027 <0.001

*^)^ For categorical data, the chi-square test or Fisher’s exact test were used, and for numerical data the *t*-test was used; seropositivity was defined as IgG antibodies ≥ 50 U/mL; seroconversion on day 28 after vaccination was defined as ≥four-fold increase compared to pre-vaccination status; and geometric mean increase (GMI) was defined as the ratio of the titer post-booster vaccination to that pre-booster vaccination.

**Table 4 vaccines-12-00344-t004:** sVNT results before and after booster dose administration.

Variables	V1		V3		*p*-Value
	Median (IQR)	*p*-Value	Median (IQR)	*p*-Value	
sVNT					
% Inhibition ChAdOX1-S (*n* = 100) CoronaVac^®^ (*n* = 99)	96.0 (94.0–96.0) 95.0 (68.0–97.0)	0.328 ^a^	96.0 (95.0–97.0) 96.0 (95.0–97.0)	0.024 ^a,^*	0.000 ^b,^* 0.000 ^b,^*
Positive proportion (*n*%) ChAdOX1-S (*n* = 100) CoronaVac^®^ (*n* = 99)	95 (95.0) 86 (86.0)	0.030 ^c,^*	100 (100) 99 (99.0)	1.000 ^d^	0.000 ^b,^* 0.000 ^b,^*

^a^ Mann–Whitney test; ^b^ Wilcoxon test; ^c^ chi-square test; ^d^ Fisher’s exact test; * significant results.

**Table 5 vaccines-12-00344-t005:** PRNT results before and after booster dose administration.

Variables	V1		V3		*p*-Value
	Median (IQR)	*p*-Value	Median (IQR)	*p*-Value	
SARS-CoV-2 Wuhan strain					
ChAdOX1-S (*n* = 8) CoronaVac^®^ (*n* = 8)	128 (20–256) 256 (64–448)	0.279 ^a^	512 (512–896) 768 (320–2048)	0.645 ^a^	0.012 ^b,^* 0.028 ^b,^*
SARS-CoV-2 Delta strain					
ChAdOX1-S (*n* = 8) CoronaVac^®^ (*n* = 8)	96 (22–128) 96 (40–416)	0.798 ^a^	384 (128–512) 256 (128–1024)	0.574 ^a^	0.051 0.091 ^b^

^a^ Mann–Whitney test; ^b^ Wilcoxon test; * significant results.

**Table 6 vaccines-12-00344-t006:** Cell-mediated immunity analysis (CD4+ and CD8+) pre- and post-booster vaccination based on vaccine priming.

Variables	V1		V3		*p*-Value ^b^
	Median (IQR)	*p*-Value ^a^	Median (IQR)	*p*-Value ^a^	
%CD4+/CD3+					
ChAdOX1-S (*n* = 24)	51.55 (33.6–70.2)	0.643	38.3 (26.7–57.7)	0.002	<0.001 *
CoronaVac^®^ (*n* = 24)	49.65 (33.8–67.8)		31.2 (22.8–58.2)		0.001 *
%IFN/CD4+					
ChAdOX1-S (*n* = 24)	0.16 (0.003–8.40)	0.014 *	3.035 (1.07–6.80)	1.00	<0.001 *
CoronaVac^®^ (*n* = 24)	0.4 (0.14–8.2)		3.205 (0.94–9.46)		0.001 *
%TNF/CD4+					
ChAdOX1-S (*n* = 24)	0.36 (0.023–6.20)	0.710	1.565 (0.82–3.06)	0.143	0.034 *
CoronaVac^®^ (*n* = 24)	0.605 (0.094–5.57)		1.23 (0.54–3.50)		0.042 *
%CD8+/CD3+					
ChAdOX1-S (*n* = 24)	9.8 (0.4–32.8)	0.959	30.45 (8.21–42.1)	0.918	0.001 *
CoronaVac^®^ (*n* = 24)	5.05 (1.3–33.3)		26.45 (12.3–41.4)		0.002 *
%IFN/CD8+					
ChAdOX1-S (*n* = 24)	1.425 (0.028–21.7)	0.184	5.93 (0.0–22.0)	0.035	<0.001 *
CoronaVac^®^ (*n* = 24)	2.515 (0.73–11.0)		8.365 (3.6–17.7)		<0.001 *
%TNF/CD8+					
ChAdOX1-S (*n* = 24)	0.36 (0.023–6.20)	0.877	4.76 (1.20–12.1)	0.001	<0.001 *
CoronaVac^®^ (*n* = 24)	0.425 (0.094–5.57)		9.775 (2.0–17.0)		<0.001 *

^a^ Mann–Whitney test; ^b^ Wilcoxon test; * significant results.

## Data Availability

Research data are available upon reasonable request to the corresponding author.

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
