# Peer review of "Immunogenicity and Safety of Half-Dose Heterologous mRNA-1273 Booster Vaccination for Adults Primed with the CoronaVac® and ChAdOx1-S Vaccines for SARS-CoV-2"

_vaccines, 2024, doi:10.3390/vaccines12040344_

Round 1

Reviewer 1 Report

Comments and Suggestions for Authors

The manuscript titled Immunogenicity and Safety of Half Dose Heterologous mRNA-1273 Booster Vaccination for Adults primed with CoronaVac® and ChAdOx1-S vaccines for SARS-CoV-2investigates the application of a half-dose booster vaccination with the mRNA-1273 COVID-19 vaccine of individuals previously vaccinated with either the CoronaVac or ChAdOx1-nCoV-19 vaccine.

 The study itself is interesting although plenty of heterologous booster vaccinations against SARS-CoV-2 have been conducted and published. Unfortunately, the poor quality of the scientific presentation, which might be partly due to linguistic problems, makes it impossible to accept the manuscript in its current form. There are a number of statements (see below), which are either incorrect or insufficiently described. Please note that due to the many cases of inconsistency of the manuscript including some discrepancy it has been impossible to address them all here. Therefore, the comments below should only be seen as examples.

 It is highly recommended that the manuscript should be subjected to a major revision, which should also include linguistics aspects reviewed by a person with good knowledge of English. 

General comments

The sentence “There was no significant difference on GMT and seropositivity rate of pre- and post-booster administration based on the vaccine priming, however a significant difference was seen after booster dose administration among the two groups” (L33-36) does not make sense and needs to be revised.

It should be stated that the “approval” for the mRNA-1273 vaccine was in the initial phase for emergency use authorization.

The original phase III trials for the BNT162b2 (Polack et al. Nature 2021, 595, 572–577) and mRNA-1273 (Baden et al. N. Engl. J. Med. 2021, 384, 403–416) are NOT anecdotal but based on real data!

The statement “with only one subject did not recover due to persistent cough within one week” (L201) is not clear. In contrast, it is mentioned in the Discussion section (L328) that the patient “still complained of cough after 28 days”. This should be mentioned in the Results section.

It is stated (L210) that “No serious adverse event was recorded”. However, in Table 2 is mentioned “Grade 4 (potentially life-threatening)”! This does not make sense!

The statement (L323) “One subject from the 3ChADoX1-S group had each back pain and low back pain” does not make sense. I assume the authors mean “both” instead of “each”.

This is impossible to understand (L357): “the imbalance between increasing angiotensin II and declined level of angiotensin 1,7”.

The statement (L395) “The results are in line with a previous study by Kang et al. on their evaluation of mRNA vaccination on cellular immunity, as inactivated-virus vaccines as CoronaVac were found to elicited higher cellular-mediated immunity as compared to mRNA vaccine” does not make any sense. In the study by Kang et al, the mRNA vaccine is compared to CoronaVac, whereas here individuals vaccinated with either CoronaVac or ChAdOx1 nCoV-19 receive the mRNA-1273 booster!

In the Conclusion section is mentioned that individuals vaccinated with the ChAdOx1 nCoV-19 vaccine had significantly higher neutralizing antibody titers and GMTs than those receiving the CoronaVac vaccines without making any references to the fact that the mean time since the vaccination was 190 days for the former and 263 days for the latter, which should influence the titers. 

Specific comments

L16: “smaller vaccine shots”? The shots are not smaller, the doses are!

L79: “after the 79 first dose was withdrawn and discarded 12 hours after the first puncture” does not make sense, please revise!

L86: “The intervention did not blind both the subjects and the investigators” is another example of poor language, although I am glad to hear that nobody lost their vision from the treatment!

L94: “who completed primary series of SARS-CoV-2 vaccine with CoronaVac®” is another example of the unacceptable poor quality of the language. I will stop here as it would take too much time to go list all the needs for revision.

L198: “hour-24” > “24 hours”

Comments on the Quality of English Language

See above!

Author Response

We have also used an English proof-reading service to elevate our manuscript. We hope the edit will suffice the critical review of the reviewer.

Response to Reviewer 1 Comments

1. Summary

2. Questions for General Evaluation

Reviewer’s Evaluation

Response and Revisions

Does the introduction provide sufficient background and include all relevant references?

Can be improved

Thank you for your evaluation, we agree some parts are hardly understood, and we have revised parts of it. The detailed revision will be enlisted below.

Are all the cited references relevant to the research?

Can be improved

Thank you for your review, we tried to use several new citations to support our study.

Is the research design appropriate?

Can be improved

We agree to the reviewers’ evaluation, and we also try to point out the limitations of this study.

Are the methods adequately described?

Must be improved

Thank you for the critical review. We tried to give several details of the methodology section.

Are the results clearly presented?

Must be improved

Thank you for this evaluation. We agree to this as we tried to point out several parts of our results, and omit some parts which might be misunderstood.

Are the conclusions supported by the results?

Must be improved

We agree to the reviewers’ evaluation. The conclusion is unclear and in need of major revision. We tried to omit some parts of the passage.

3. Point-by-point response to Comments and Suggestions for Authors

Comments 1: The sentence “There was no significant difference on GMT and seropositivity rate of pre- and post-booster administration based on the vaccine priming, however a significant difference was seen after booster dose administration among the two groups” (L33-36) does not make sense and needs to be revised.

Response 1: Thank you for pointing this out. We tried to explain that there was a significant difference between the two groups and also a difference between the pre-booster and post-booster stages. However, we realized it is confusing and thus revised it accordingly by omitting the line “pre- and post-booster administration based on the vaccine priming, however a significant difference was seen after” in L35-36.

Comment 2: It should be stated that the “approval” for the mRNA-1273 vaccine was in the initial phase for emergency use authorization.

Response 2: Thank you for the suggestion. We inserted this statement L49 of the introduction section.

Comment 3: The original phase III trials for the BNT162b2 (Polack et al. Nature 2021, 595, 572–577) and mRNA-1273 (Baden et al. N. Engl. J. Med. 2021, 384, 403–416) are NOT anecdotal but based on real data!

Response 3: We are aware of the misused word on the passage. We have revised the passage “several anedoctal reports” to “numerous reports” in L51 based on the reviewers’ suggestion.

Comment 4: The statement “with only one subject did not recover due to persistent cough within one week” (L201) is not clear. In contrast, it is mentioned in the Discussion section (L328) that the patient “still complained of cough after 28 days”. This should be mentioned in the Results section.

Response 4: Thank you for the suggestion. We have added the words “after 28 days of observation” in L212 in the Results section.

Comment 5: It is stated (L210) that “No serious adverse event was recorded”. However, in Table 2 is mentioned “Grade 4 (potentially life-threatening)”! This does not make sense!

Response 5: We acknowledge our misleading words in the passage. We used two different classifications of adverse events, using the FDA classification into grade 1-4, and another classification by the GCP guideline. As one of the patients complained of fever >40oC, the patient is classified into the grade 3 of FDA classification (as seen in Table 2) after a thorough discussion. On the other hand, despite the persistent fever, there is no death or disability caused by the vaccine, thus there is no serious adverse event recorded. We revised the adverse event, which previously classified as grade 4 into grade 3. We hope the revision will suffice to correct the passage.

Comment 6: The statement (L323) “One subject from the 3ChADoX1-S group had each back pain and low back pain” does not make sense. I assume the authors mean “both” instead of “each”.
Response 6:
Thank you for the suggestion. We revised the line accordingly.

Comment 7: This is impossible to understand (L357): “the imbalance between increasing angiotensin II and declined level of angiotensin 1,7”.

Response 7: Thank you for the critical review. We revised the passage to “the imbalance of angiotensin level” L377.

Comment 8: The statement (L395) “The results are in line with a previous study by Kang et al. on their evaluation of mRNA vaccination on cellular immunity, as inactivated-virus vaccines as CoronaVac were found to elicited higher cellular-mediated immunity as compared to mRNA vaccine” does not make any sense. In the study by Kang et al, the mRNA vaccine is compared to CoronaVac, whereas here individuals vaccinated with either CoronaVac or ChAdOx1 nCoV-19 receive the mRNA-1273 booster!

Response 8: We are grateful for the critical review on our discussion. On this passage, we would like to clarify that the line implied that similar results by Kang et al. was on the comparison of CoronaVac and mRNA vaccine on their cellular immunity regardless of the mRNA-1273 booster dose. We believe the results by Kang et al. is still relevant to support our claim. Thus, we decided to keep the line as it is. However, we are open for another discussion and consider omitting or revise the line should the reviewer suggest to do so.

Comment 9: In the Conclusion section is mentioned that individuals vaccinated with the ChAdOx1 nCoV-19 vaccine had significantly higher neutralizing antibody titers and GMTs than those receiving the CoronaVac vaccines without making any references to the fact that the mean time since the vaccination was 190 days for the former and 263 days for the latter, which should influence the titers.

Response 9: Thank you for the critical review. We realized the risk of data bias due to the significant difference of time-to-vaccination between the two group. The difference was caused by the difference of vaccine approval, thus most of the CoronaVac subjects were vaccinated early. However, all the subjects were deemed eligible based on the inclusion criteria and thus, despite the difference, we push through the trial to determine the need for booster dose amidst the increasing Omicron case. We also decided to omit the part in the conclusion section to avoid confusion. We are open for further discussion.

4. Response to Comments on the Quality of English Language

Point 1: L16: “smaller vaccine shots”? The shots are not smaller, the doses are!

Response 1: Thank you for the suggestion. We have revised it accordingly.

Point 2: L79: “after the 79 first dose was withdrawn and discarded 12 hours after the first puncture” does not make sense, please revise!

Response 2: Thank you for the suggestion. We decided to revise the passage into Vials were stored between 2oC to 25oC during usage and discarded 12 hours after the first time used for vaccination” which can be seen in L82-83.

Point 3: L86: “The intervention did not blind both the subjects and the investigators” is another example of poor language, although I am glad to hear that nobody lost their vision from the treatment!

Response 3: We are aware of the poor use of language in this passage. We revised the line into “There were no blinding during the intervention on both subjects and investigators during the booster injection” in L89-90. We hope this revision is clear for the readers.

Point 4: L94: “who completed primary series of SARS-CoV-2 vaccine with CoronaVac®” is another example of the unacceptable poor quality of the language. I will stop here as it would take too much time to go list all the needs for revision.

Response 4: Thank you for the review. We decided to add the word “dose” in L99.

Point 5: L198: “hour-24” > “24 hours”

Response 5: Thank you for the suggestion. We have revised it accordingly.

5. Additional clarifications

We are grateful to the review and suggestion by Reviewer 1. We realized the limited use of language in our manuscript, thus we decided to use a proof-reading service to elevate our manuscript. We hope this revision will suffice for our manuscript to be considered for acceptance.

Reviewer 2 Report

Comments and Suggestions for Authors

The aim of the study on the Immunogenicity and Safety of Half-Dose Heterologous mRNA-1273 Booster Vaccination for Adults primed with CoronaVac® and ChAdOx1-S vaccines for SARS-CoV-2 was to evaluate the effectiveness and safety of administering a half-dose mRNA-1273 booster vaccine to adults who had previously received different primary vaccines (CoronaVac® and ChAdOx1-S) for SARS-CoV-2.

Since the aim is to use a half dose of the mRNA-1273 vaccine on people primed with different vaccines, there should be a comparative group that receives the full dose of the mRNA vaccine to evaluate the effectiveness of the half dose. Unfortunately, the study did not include this study group, and it is impossible to conduct such a study based on this well-written draft. I could suggest the authors include this as a limitation of the study and discuss it in details with several clinical studies conducted before on the effectiveness of the half dose on the immune response. There are minor comments below:

Please correct the temperature degree symbol "°" in the text.

Add a period to the end of the sentence at line 202.

Please use either "ChAdOx1-S" or "AstraZeneca" consistently throughout the draft, figures, and tables.

Could the authors create a graph showing cellular immune responses instead of Table 6?

Please provide the full name of "AE" where it was used for the first time.

Author Response

Response to Reviewer 2 Comments

1. Summary

2. Questions for General Evaluation

Reviewer’s Evaluation

Response and Revisions

Does the introduction provide sufficient background and include all relevant references?

Yes

Thank you for your evaluation. We are grateful for the positive review in this aspect.

Are all the cited references relevant to the research?

Yes

Thank you for your evaluation. We are grateful for the positive review in this aspect.

Is the research design appropriate?

Can be improved

We agree to the reviewers’ evaluation, and we also try to point out the limitations of this study.

Are the methods adequately described?

Yes

Thank you for your evaluation. We are grateful for the positive review in this aspect.

Are the results clearly presented?

Can be improved

Thank you for this evaluation. We agree to this as we tried to point out several parts of our results and omit some parts which might be misunderstood.

Are the conclusions supported by the results?

Can be improved

We agree to the reviewers’ evaluation and decided to improve the conclusion based on the reviewers’ suggestion.

3. Point-by-point response to Comments and Suggestions for Authors

Comment 1: The aim of the study on the Immunogenicity and Safety of Half-Dose Heterologous mRNA-1273 Booster Vaccination for Adults primed with CoronaVac® and ChAdOx1-S vaccines for SARS-CoV-2 was to evaluate the effectiveness and safety of administering a half-dose mRNA-1273 booster vaccine to adults who had previously received different primary vaccines (CoronaVac® and ChAdOx1-S) for SARS-CoV-2.

Since the aim is to use a half dose of the mRNA-1273 vaccine on people primed with different vaccines, there should be a comparative group that receives the full dose of the mRNA vaccine to evaluate the effectiveness of the half dose. Unfortunately, the study did not include this study group, and it is impossible to conduct such a study based on this well-written draft. I could suggest the authors include this as a limitation of the study and discuss it in details with several clinical studies conducted before on the effectiveness of the half dose on the immune response.

Response 1: Thank you for the critical review by the reviewer. We acknowledge the limitation of our study with no comparator of the full-dose group to determine the non-inferior effect of the half-dose booster vaccine. Our study is supported by other reports evaluating the comparison between half-dose vs full-dose booster vaccine in BNT162b2 (Pfizer-BioNTech) vaccine. Another study was conducted in Indonesia prior to this study using the mRNA-1273 vaccine. Thus, we believe the use of half-dose booster vaccine can be justified with such reports. We added this to the limitation section in the discussion section. We are open for further discussion.

Comment 2: Please correct the temperature degree symbol "°" in the text.

Response 2: Thank you for the suggestion. We have revised it accordingly.

Comment 3: Add a period to the end of the sentence at line 202.

Response 3: Thank you for the suggestion. We have revised it accordingly.

Comment 4: Please use either "ChAdOx1-S" or "AstraZeneca" consistently throughout the draft, figures, and tables.

Response 4: Thank you for the suggestion. We have revised it accordingly.

Comment 5: Could the authors create a graph showing cellular immune responses instead of Table 6?

Response 5: Thank you for the suggestion. We agree the table format will make it difficult for readers to take highlights of the study. We have added new figure(s) or graph(s) as seen in Figure 4.

Comment 6: Please provide the full name of "AE" where it was used for the first time.

Response 6: Thank you for the suggestion. We have revised it accordingly.

4. Response to Comments on the Quality of English Language

None.

5. Additional clarifications

We are grateful for the positive feedback by Reviewer 2. We have added the concerning issues on the non-existent of full-dose comparator. However, we hope our explanation and addition in the discussion section suffice the reviewer.

Reviewer 3 Report

Comments and Suggestions for Authors

Putri et al. conducted an observational stud comparing the boosting effect of Moderna mRNA vaccine after prime-vaccination with CoronaVac® and ChAdOx1-S vaccines. The authors enrolled a total of 200 participants and monitored them for one month after administering the half-dose of the Moderna vaccine.

The article is well written, the results clearly presented, and the conclusions summarize the results presented. However, some modifications are required before publication.

-Please provide further details for the exclusion criteria for COVID-19 confirmed cases.

-Were the participants monitored for COVID-19 infections during the study? Could the authors measure N-antibodies to exclude participants that were infected during that time?

-Figure 2 and Figure 3 please update the graph.

·        Add Y-axes units, legends and increase the size of both axes.

·        Also include the statistical comparison to Figure 2.

·        Please harmonize formats.

-I cannot find the data in Table 3 cited in lines 220-221.

-Please include the gating strategy of the ICS analysis.

-Why only n=8 were used in PRNT, how relevant is that data?

-Ancestral SARS-CoV-2 was not circulating at the time of the study. Could the authors provide evidence of cross-reactive immune response against Omicron BA.1 or current VOCs?

-How does the antibody binding titers and neutralizing antibody titers obtained in this study compare with a booster with a total dose of mRNA? Could we reduce even more the dose of the booster?

-Will the authors check the longevity if this immune response after 28 days?

Author Response

Thank you for the critical review. 

Response to Reviewer 3 Comments

1. Summary

2. Questions for General Evaluation

Reviewer’s Evaluation

Response and Revisions

Does the introduction provide sufficient background and include all relevant references?

Yes

Thank you for your evaluation. We are grateful for the positive review in this aspect.

Are all the cited references relevant to the research?

Yes

Thank you for your evaluation. We are grateful for the positive review in this aspect.

Is the research design appropriate?

Can be improved

We agree to the reviewers’ evaluation, and we also try to point out the limitations of this study.

Are the methods adequately described?

Can be improved

Thank you for the critical review. We tried to give several details of the methodology section of this study and add more details on the ICS analysis.

Are the results clearly presented?

Can be improved

Thank you for this evaluation. We agree to this as we tried to point out several parts of our results and omit some parts which might be misunderstood.

Are the conclusions supported by the results?

Can be improved

Thank you for the critical review. We have tried to revise the conclusion.

3. Point-by-point response to Comments and Suggestions for Authors

Comment 1: Putri et al. conducted an observational stud comparing the boosting effect of Moderna mRNA vaccine after prime-vaccination with CoronaVac® and ChAdOx1-S vaccines. The authors enrolled a total of 200 participants and monitored them for one month after administering the half-dose of the Moderna vaccine. The article is well written, the results clearly presented, and the conclusions summarize the results presented. However, some modifications are required before publication.

Response 1: Thank you for positive review of our manuscript. We hope some revisions below will suffice the reviewers’ suggestion.

Comment 2: Please provide further details for the exclusion criteria for COVID-19 confirmed cases.

Response 2: Subjects with confirmed COVID-19 infection were evaluated based on the laboratory results 1 month prior to the recruitment. The addition has been made in L110 of the Methods section.

Comment 3: Were the participants monitored for COVID-19 infections during the study? Could the authors measure N-antibodies to exclude participants that were infected during that time?

Response 3: We have excluded subjects with laboratory confirmed COVID-19 infection and monitoring for the subjects on COVID-19 infection was conducted based on the symptoms suggestive of COVID-19. We conducted a nasopharyngeal swab on these subjects. However, no N-antibodies measurement was conducted to exclude such patients.

Comment 4: Figure 2 and Figure 3 please update the graph.

·        Add Y-axes units, legends and increase the size of both axes.

·        Also include the statistical comparison to Figure 2.

·        Please harmonize formats.

Response 4: Thank you for the critical review. We have revised figure 2 to harmonize the formats and add details on statistical analysis. We hope the revision will suffice the reviewers’ comment.

Comment 5: I cannot find the data in Table 3 cited in lines 220-221.

Response 5: We are aware of such mistakes. We have decided to omit the passage as it is not relevant. The data on sVNT can be seen in table 4.

Comment 6: Please include the gating strategy of the ICS analysis.

Response 6: Thank you fro the suggestion. We have added the passage on the gating strategy on L181-183. However, the full gating strategy will be presented in the supplement as we try to limit the results and discussion.

Comment 7: Why only n=8 were used in PRNT, how relevant is that data?

Response 7: We are aware that the use of only 8 subjects in the analysis is not sufficient in having an objective output. We decided to only include 8 subjects due to the limitation of budget. However, we still believe the results is still laid out to support our report on the inihibitory capacity of the vaccine on both Wuhan strain (wild-type) and Delta-strain.

Comment 8: Ancestral SARS-CoV-2 was not circulating at the time of the study. Could the authors provide evidence of cross-reactive immune response against Omicron BA.1 or current VOCs?

Response 8: Thank you for the critical comment. We don’t conduct analysis on the COVID-19 strain circulating in our study partly due to the non-available Omicron PRNT kit at the time of the study. However, data on the Jakarta circulating SARS-CoV-2 reveals the dominant presence of Omicron variant. In short term duration of observation during our study. There were no case of COVID-19 in our subjects which might answer the possibility of cross-reactive immune response to the Omicron variant.

Comment 9: How does the antibody binding titers and neutralizing antibody titers obtained in this study compare with a booster with a total dose of mRNA? Could we reduce even more the dose of the booster?

Response 9: Thank you for the critical question. Unfortunately, we do not conduct the antibody binding titers analysis compared to the full dose vaccine. We have added this as a limitation on the discussion section in our manuscript. We hope this will suffice to answer the question by the reviewer.  

Comment 10: Will the authors check the longevity if this immune response after 28 days?

Response 10: Thank you for the critical question. Unfortunately, we do not assess the immune response after 28 days in our subjects. This might be a critical point for further studies concerning the use of half-dose mRNA vaccine.

4. Response to Comments on the Quality of English Language

None.

5. Additional clarifications

We offer our utmost gratitude for the positive feedback and critical review in our manuscript. We might not be able to include some analysis suggested by Reviewer 3, however we hope our answers and suggestion suffice the reviewers’ comment.

Round 2

Reviewer 1 Report

Comments and Suggestions for Authors

The authors have addressed the points I raised successfully. They have also revised the manuscript according to the comments made by the other reviewers.

Reviewer 2 Report

Comments and Suggestions for Authors

Thanks for addressing my concerns in the revised version. I've found that this study can fill the gap in the field.